# An NKX-COUP-TFII morphogenetic code directs mucosal endothelial addressin expression

Thanh Theresa Dinh[1,2,15], Menglan Xiang[1,2,15], Anusha Rajaraman[1,2,3,15], Yongzhi Wang[1,4], Nicole Salazar[1], Yu Zhu[1], Walter Roper[5], Siyeon Rhee[6], Kevin Brulois[1,2], Ed O'Hara[2], Helena Kiefel[1], Truc M. Dinh[2], Yuhan Bi[1,2], Dalila Gonzalez[7], Evan P. Bao[2], Kristy Red-Horse[6,8,9], Peter Balogh[10,11], Fanni Gábris[10,11], Balázs Gaszner[12], Gergely Berta[13], Junliang Pan[2,14] ✉ & Eugene C. Butcher[1,2,14] ✉

Immunoglobulin family and carbohydrate vascular addressins encoded by *Madcam1* and *St6gal1* control lymphocyte homing into intestinal tissues, regulating immunity and inflammation. The addressins are developmentally programmed to decorate endothelial cells lining gut post-capillary and high endothelial venules (HEV), providing a prototypical example of organ- and segment-specific endothelial specialization. We identify conserved NKX-COUP-TFII composite elements (NCCE) in regulatory regions of *Madcam1* and *St6gal1* that bind intestinal homeodomain protein NKX2-3 cooperatively with venous nuclear receptor COUP-TFII to activate transcription. The *Madcam1* element also integrates repressive signals from arterial/capillary Notch effectors. Pan-endothelial COUP-TFII overexpression induces ectopic addressin expression in NKX2-3[+] capillaries, while NKX2-3 deficiency abrogates expression by HEV. Phylogenetically conserved NCCE are enriched in genes involved in neuron migration and morphogenesis of the heart, kidney, pancreas and other organs. Our results define an NKX-COUP-TFII morphogenetic code that targets expression of mucosal vascular addressins.

The blood vasculature is segmentally specialized for physiological functions specific to arterial, capillary, venous, and sinusoidal vessels, and regionally specialized for functions specific to different organs, tissues, or microenvironments[1]. A prototypical example is the segmental and organotypic specialization of vascular endothelium for control of lymphocyte and immune cell traffic[2]. Blood-borne leukocytes adhere and extravasate preferentially through vascular segments immediately downstream of capillaries: postcapillary venules (PCV) in

[1]Laboratory of Immunology and Vascular Biology, Department of Pathology, Stanford University School of Medicine, Stanford, CA, USA. [2]Palo Alto Veterans Institute for Research, Palo Alto, CA, USA. [3]Department of Molecular Cell Biology and Immunology, Vrije Universiteit Medical Center, Amsterdam, The Netherlands. [4]Department of Clinical Science Malmo, Section of Surgery, Lund University, Malmo, Sweden. [5]Columbia University Vagelos College of Physicians and Surgeons, New York City, NY, USA. [6]Department of Biology, Stanford University, Stanford, CA, USA. [7]University of California, San Diego, La Jolla, CA, USA. [8]Institute for Stem Cell Biology and Regenerative Medicine, Stanford University School of Medicine, Stanford, CA, USA. [9]Howard Hughes Medical Institute, Stanford, CA, USA. [10]Department of Immunology and Biotechnology, University of Pécs Medical School, Pécs, Hungary. [11]Lymphoid Organogenesis Research Team, Szentágothai Research Center, Pécs, Hungary. [12]Department of Anatomy, University of Pécs Medical School, Pécs, Hungary. [13]Department of Medical Biology and Central Electron Microscopy Laboratory, University of Pécs Medical School, Pécs, Hungary. [14]The Center for Molecular Biology and Medicine, Veterans Affairs Palo Alto Health Care System, Palo Alto, CA, USA. [15]These authors contributed equally: Thanh Theresa Dinh, Menglan Xiang, Anusha Rajaraman. ✉e-mail: jpan@stanford.edu; ebutcher@stanford.edu

sites of inflammation and immune surveillance, and specialized high endothelial venules (HEV) in secondary and tertiary lymphoid tissues[3–6]. HEV and PCV in sites of immune surveillance display organ-specific adhesion molecules termed vascular addressins that direct tissue-specific lymphocyte homing in support of regional immune specialization and responses[2,3,7]. The mucosal vascular addressin-1, MAdCAM1, is an Ig and mucin family member expressed by HEV in gut-associated lymphoid tissues (GALT) and by venules for lymphocyte homing into the intestinal lamina propria, and by the splenic marginal zone; but not normally by HEV in other tissues in the adult[8,9]. MAd-CAM1 recruits lymphocytes bearing the gut-homing receptor integrin $\alpha_4\beta_7$. Intestinal HEV and PCV are also decorated by α2-6 sialic acid-capped glycotopes recognized by the B cell-specific lectin CD22 (Siglec2)[10]. These glycotopes constitute a B cell-specific mucosal vascular addressin, designated here BMAd, which selectively enhances the homing of B cells into GALT[10], where they contribute to the IgA response to intestinal antigens and pathogens[11]. BMAd is synthesized by beta-galactoside alpha-2,6-sialyltransferase 1, encoded by *St6gal1*, which like *Madcam1* is selectively expressed by GALT HEV[10].

Whole genome expression studies have shown that developmentally important transcriptional programs are retained and expressed basally in the mature vascular endothelium[10,12,13]. Venular endothelial cells express *Nr2f2*-encoding COUP-TFII, a master transcription factor for embryonic vein formation[14–16]. Capillary and arterial EC display Notch and its downstream repressor-encoding genes including *Hey1* and *Hes1*[10,12], critical participants in arterial specification. Counterplay between COUP-TFII and Notch signaling ensures separation of the unique features of the vascular segments[17,18]. Intestinal EC retains expression of *Nkx2-3*, encoding a homeodomain transcription factor that programs gut development[19]. Moreover, COUP-TFII and proteins of the NKX2 family cooperate to regulate cell fate determination during development. For example, the embryonic heart field is established by NKX2-5, the vertebrate homolog of Drosophila "tinman"[20]; within the heart field COUP-TFII programs atrial fate[15]. COUP-TFII and NKX2-5 also control pro-epicardial and endocardial development[15,21], while COUP-TFII and NKX2-1 coordinate to direct the migration of developing GABAergic neurons[22–24]. These observations raised the possibility that cooperative interplay of NKX2-3 and COUP-TFII might contribute to organotypic specialization of intestinal venules. While *cis*-acting composite motifs that are implicated in endothelial-specific gene expression during early development have been defined[25], genomic motifs and transcriptional control mechanisms that direct organotypic and segmental specialization of endothelium, including the venule- and intestine-specific expression of the addressins, remain to be elucidated.

Here we describe NKX-COUP-TFII composite elements (NCCE), short genomic sequences that direct organ- and segment-specific expression of the mucosal vascular addressins. NCCE in the *Madcam1* and *St6gal1* regulatory regions seed cooperative binding of the intestinal homeodomain transcription factor (TF) NKX2-3 with the venous master regulator COUP-TFII, integrating regional and segmental factors to create a transcriptional activation complex that promotes addressin expression. Phylogenetically conserved NCCE genomic motifs are enriched in association with genes involved in embryonic organ morphogenesis, including genes implicated in cardiovascular, pancreatic, and neuronal specification. Thus our results show that COUP-TFII forms transcriptional activation complexes with NKX homeodomain partners on NCCE to target the vascular addressins, and suggest roles of NCCE in embryonic organogenesis and cell specification.

## Results

### A *Madcam1* composite element (CE) binds gut, vein, and capillary-specific TF

We identified a combinatorial element (CE) within the promotor of *Madcam1* comprising 1) a homeodomain (HD) motif that binds the intestinal TF NKX2-3; 2) two canonical binding sites for the venous TF COUP-TFII (encoded by *Nr2f2*); and 3) a conserved E-box that binds the capillary/arterial Notch downstream transcriptional repressor HEY1 (Figs. 1A and S1A). The two COUP-TFII "half sites" are spaced seven nucleotides apart in the mouse (TGACCC, highlighted in green in Figs. 1A and S1A), a spacing consistent with COUP-TFII homodimer binding[26]. The HD, E-box and COUP-TFII "A" site motifs overlap (Fig. 1A), suggesting the potential for competition between TFs.

To test for direct binding of the TFs at the *Madcam1* CE, we performed electromobility shift assays (EMSA) using recombinant proteins and a labeled *Madcam1* CE probe (Fig. S1A). Binding of HEY1, NKX2-3, and COUP-TFII to the *Madcam1* CE probe was confirmed by supershift of the TF:CE complexes with respective anti-TF antibodies (Fig. S1A, lanes 3, 8, and 13). NKX2-3:CE binding was abolished by a mutation of the homeodomain site (Fig. S1A, lane 10); and HEY1:CE binding was abolished by a mutation of the E-box motif (Fig. S1A, lane 5). In addition, COUP-TFII:CE binding was blocked by mutation of the two COUP-TFII binding sites (Fig. S1A; lane 15). These results confirm that the identified motifs mediate TF:CE binding. Moreover, as predicted, HEY1 dose-dependently inhibited NKX2-3 binding to the "A" site, indicating that binding to the overlapping motifs is competitive (Fig. S1B).

### The CE integrates NKX2-3, COUP-TFII, and HEY1 to regulate expression

We used luciferase reporter assays to assess the role of the CE in regulating expression from the *Madcam1* promoter in 293T cells. We transiently transfected wild-type or mutant CE-containing reporter constructs with or without co-transfected *Nkx2-3, Nr2f2*, or *Hey1* and assessed luciferase expression in the presence of TNFα to drive expression from proximal NFκB sites[27]. *Nkx2-3* overexpression enhanced transcription from the wild-type reporter (Fig. 1B). Mutation of the NKX2-3 binding site in the CE abrogated expression, reducing reporter activity to below the control without *Nkx2-3* overexpression (Fig. 1B). Overexpression of *Nr2f2* alone did not activate the reporter, but *Nr2f2* cooperated with *Nkx2-3* to promote enhancer activity (Fig. 1B and C). Reporter activation by *Nkx2-3* or by *Nkx2-3* and *Nr2f2* was inhibited dose-dependently by *Hey1* (Fig. 1C), consistent with TF competition (Figure S1B).

To address CE function independently of proximal sequences, we utilized a reporter consisting of three tandem CE motifs driving a minimal promoter. Mutational analyses showed that activation of reporter expression in 293T cells by *Nr2f2* and *Nkx2-3* required the COUP-TFII "B" binding site, whereas COUP-TFII "A" site mutation had little effect (Fig. 1D), suggesting recruitment of an activating COUP-TFII:NKX2-3 complex on the CE. We conclude that the *Madcam1* CE integrates cooperative transcriptional activation by *Nkx2-3* and *Nr2f2*, in competition with the Notch downstream repressor *Hey1*.

To assess the regulation of the endogenous endothelial *Madcam1* gene we used bEnd.3 cells, a mouse endothelial cell line that supports TNFα-induced *Madcam1* expression[27,28]. bEnd.3 express low levels of *Nkx2-3*, *Nr2f2*, and *Hey1* (raw expression values: 324, 1495, 295, respectively; see Methods). Overexpression of *Nkx2-3* enhanced *Madcam1* expression (Fig. 1E). shRNAs targeting *Nkx2-3* or *Nr2f2* in bEnd.3 reduced endogenous *Madcam1* expression in the presence of TNF (Fig. 1F). Thus, both NKX2-3 and COUP-TFII are required for endogenous *Madcam1* expression in endothelial cells.

To address the possibility that endogenous Notch signaling moderates bEnd.3 responses, we stably transduced bEnd.3 with dominant negative *MAML* (*DN-MAML*), a mutant *Mastermind-like 1* gene which potently inhibits Notch-dependent expression of HEY1 and HES1[29]. Induction of *Madcam1* and of surface MAdCAM1 protein was dramatically enhanced in DN-MAML EC compared to the parent line (Fig. 1G and inset), and overexpression of *Hey1* suppressed MAdCAM1 induction (Fig. 1G).

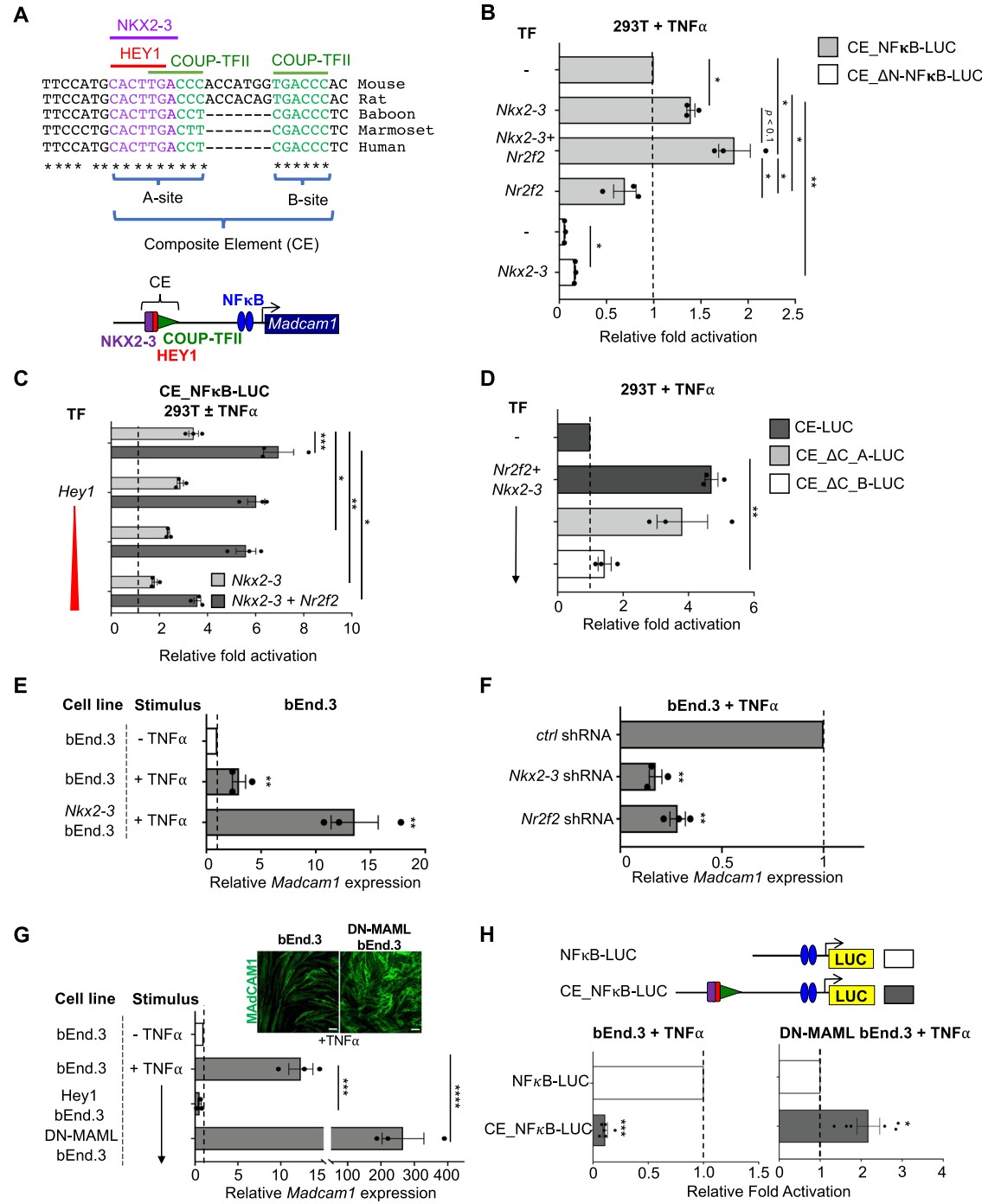

**Fig. 1 | An evolutionarily conserved composite element (CE) in the *Madcam1* promoter integrates NKX2-3, COUP-TFII, and HEY1 to regulate transcription.** **A** Sequence of the CE in indicated species, and schematic of the CE. Conserved E-box (HEY1 binding site), homeodomain (NKX2-3 binding site) and COUP-TFII binding sites are highlighted in red, purple and green respectively. **B** *Nkx2-3* or cooperative *Nkx2-3* and *Nr2f2* enhance TNFα stimulated luciferase reporter activity driven by CE-containing *Madcam1* promoter (CE_NFκB-LUC) in 293T cells. Mutation of the homeodomain motif (in CE_ΔN-NFκB-LUC) inhibits activity. Data normalized to CE_NFκB-LUC activity. **C** HEY1 dose-dependently suppresses *Nkx2-3* (light gray bars) or cooperative *Nr2f2* and *Nkx2-3* (dark gray bars) transcriptional activation from the CE-containing *Madcam1* promoter in TNFα stimulated 293T cells. Data normalized to basal reporter activity without TNFα stimulation (dashed line). **D** Effects of mutation of COUP-TFII "A" (CE_ΔC_A-LUC) or "B" (CE_ΔC_B-LUC) sites on cooperative *Nr2f2* and *Nkx2-3* activation of luciferase reporter in 293T cells. Results were normalized to control CE-containing reporter activity. Vector contains three tandem copies of CE. **E** Induction of *Madcam1* expression in TNFα stimulated

bEnd.3 cells overexpressing *Nkx2-3*, evaluated by real-time PCR. Results are normalized to basal *Madcam1* levels in bEnd.3 cells. **F** Knockdown of *Nkx2-3* or *Nr2f2* by shRNA inhibits *Madcam1* expression in bEnd.3 cells. Data were normalized to β-actin and then to expression in cells transduced with control shRNA. **G** Endogenous *Madcam1* expression in bEnd.3 cells or bEnd.3 cells stably transduced with *Hey1* or *DN-MAML*, evaluated by real-time PCR. Results are normalized to basal *Madcam1* in unstimulated bEnd.3 cells. Inset: immunofluorescence staining showing MAdCAM1 expression (green) in TNFα-stimulated bEnd.3 vs DN-MAML bEnd.3 cells. Scale bars: 20μm. **H** Upper panel: schematic of the *Madcam1* promoter with (CE_NFκB-LUC) or without (NFκB-LUC) the CE. Lower panel: luciferase reporter activity driven by the *Madcam1* promoter with (CE_NFκB-LUC; gray bars) or without (NFκB-LUC; white bars) the CE. CE_NFκB-LUC activity is normalized to the activity of NFκB-LUC. All TNFα-stimulations were done for 36 h. Values are mean ± SEM of three independent experiments unless stated otherwise. *: *p* value < 0.05; **: *p* value < 0.01; ***: *p* value < 0.005; ****: *p* value < 0.001. Two tailed t-test, paired.

To assess the function of the *Madcam1* CE in endothelial cells, we generated luciferase (LUC) reporter constructs driven by the proximal NF-κB site with or without the CE, and transfected the constructs into bEnd.3 cells, (Fig. 1H; sequence information in Supplementary Table S1). The proximal NF-κB site by itself drove robust TNFα-responsive luciferase expression. However, promoter activity was dramatically inhibited by inclusion of the CE (Fig. 1H), suggesting that the CE engages inhibitory signals active in bEnd.3 cells. We reasoned that E-box binding transcriptional repressors, which are induced by Notch signaling, might mediate this transcriptional inhibition. To assess the function of the CE in the absence of Notch signaling, we transfected the reporter constructs into DN-MAML cells. In contrast to its inhibitory effect in bEnd.3 EC, the CE-containing reporter construct significantly enhanced LUC activity in DN-MAML EC (Fig. 1H). Thus Notch signaling via its downstream repressors inhibits both CE activity and endogenous *Madcam1* induction in endothelial cells.

Although the spacing and sequence of TF binding sites in the human *MADCAM1* CE differ from those in the mouse (Fig. 1A), reporter assays confirmed conservation of CE regulatory functions including activation by co-transfection of *NKX2-3* and *NR2F2* (Fig. S2A) and competitive inhibition by *HEY1* (Fig. S2B). Tandem *MADCAM1* CE sequences without the proximal NFκB site were sufficient to confer reporter activity from a minimal promoter in response to *NKX2-3* and *NR2F2*, as well as negative regulation by *HEY1* (Fig. S2C). These results support functional conservation of the regulatory interplay of NKX2-3, COUP-TFII, and HEY1 for combinatorial control of *MADCAM1* transcription.

## NKX2-3 and COUP-TFII form DNA- and tinman-dependent heterodimers

Coactivation of the *Madcam1* CE-containing reporter by COUP-TFII and NKX2-3 suggested coordinate binding of these TFs to the CE. To address this possibility, we used EMSA to assess the interaction of NKX2-3 and COUP-TFII on the native and mutant CE probes (Fig. 2A). Co-incubation of NKX2-3 and COUP-TFII with native CE yielded a band distinct from those formed by CE bound to NKX2-3 or COUP-TFII alone (Fig. 2B, left, blue arrow). This band was intensified by mutation of the COUP-TFII "A" site (Fig. 2B, right, lane 8), possibly reflecting competition of COUP-TFII and NKX2-3 for "A" site binding. Supershift with an NKX2-3 antibody (Fig. 2B, right, red star) but not by control antibody (Fig. 2B, right, lane 10) confirmed NKX2-3 participation in the complex. When we abolished both the NKX2-3 and COUP-TFII "A" binding sites, NKX2-3 no longer bound to the probe (Fig. 2C, lane 2); but it retained the ability to bind together with COUP-TFII to form an NKX2-3:COUP-TFII:DNA complex (Fig. 2C, lane 3, blue arrow; lane 4, red star). Formation of the heterotrimeric NKX2-3:COUP-TFII:DNA complex even in the absence of a canonical NKX2-3 binding site suggests that COUP-TFII, bound to DNA, can initiate formation of the gene regulatory complex through protein–protein interactions, which could then be stabilized by the adjacent homeodomain motif. Binding of COUP-TFII to the endogenous *Madcam1* NCCE in DN-MAML bEnd.3 cells were confirmed by ChIP-PCR (Fig. S3).

NKX2-3 contains an N-terminal tinman (TN) domain, conserved in members of the NKX2 family. Recombinant NKX2-3 lacking the TN domain failed to dimerize with COUP-TFII on CE (Fig. 2C, lane 6 vs 3), suggesting a role for the domain in NKX2-3:COUP-TFII association. Consistent with this, a synthetic 24-amino acid peptide encompassing the TN domain competitively inhibited NKX2-3:COUP-TFII heterodimerization, while a scrambled peptide had no effect (competitive EMSA, Fig. 2D). Moreover, transduction with a retroviral construct encoding the TN domain peptide suppressed basal *Madcam1* and abrogated the enhanced *Madcam1* induction seen in DN-MAML bEnd.3 cells transfected with NKX2-3 (Fig. 2E). Thus, NKX2-3:COUP-TFII complex formation and function are dependent on physical interactions involving the NKX2-3 TN domain. Proximity ligation assay

(PLA) confirmed the physical association of NKX2-3 and COUP-TFII within the nuclei of EC expressing MAdCAM1 (Fig. 2F).

## Identification of a conserved *St6gal1* NCCE

*St6gal1* confers expression of the B cell-recruiting glyco-addressin BMAd, and like *Madcam1* is selectively expressed by intestinal HEV[10]. We identified a conserved NKX-COUP-TFII composite element (NCCE) within an enhancer region in the second intron of *St6gal1* (Fig. 3A), suggesting a common genomic address code. NKX2-3 and COUP-TFII both bound the native *St6gal1* NCCE (Fig. 3B), and binding was inhibited by cold wild type probe but not by mutant probes lacking the cognate homeodomain or COUP-TFII motifs (Fig. S4). Co-incubation with NKX2-3 and COUP-TFII produced a distinct complex (Fig. 3B, blue arrow), indicative of heterodimer formation as seen with the *Madcam1* CE (Fig. 2B). Endogenous expression of *St6gal1* in DN-MAML bEnd.3 cells was inhibited by shRNA knockdown of *Nr2f2* or *Nkx2-3* (Fig. 3C) and conversely upregulated by overexpressing *Nkx2-3* (Fig. 3D). *St6gal1* was suppressed by overexpression of the NKX2-3 tinman domain (Fig. 3D), supporting the requirement for both TFs and for TN-dependent interactions for optimal *St6gal1* expression in EC. Tandem *St6gal1* NCCE sequences activated a minimal luciferase reporter in response to transfected COUP-TFII or NKX2-3, and co-transfection of both TFs further enhanced activation (Fig. 3E, gray bars). Mutation of the COUP-TFII site suppressed the cooperative response (Fig. 3E, white bars). Thus the intronic NCCE in *St6gal1*, like that in the *Madcam1* promoter, is sufficient to confer combinatorial activation of gene expression by NKX2-3 and COUP-TFII.

## NKX2-3 is required for BMAd expression in Peyer's patch HEV

The identification of the *St6gal1* NCCE suggested that NKX2-3 might drive BMAd expression in gut venules. To address this hypothesis, we used the *Sambucus nigra* lectin (SNA) to assess the vascular display of alpha-2,6-sialic acid glycotopes. Perfused SNA did not stain EC in *St6gal1*[−/−] mice, confirming the selectivity for BMAd glycotopes (Fig. S5). SNA stained Peyer's patch (PP) HEV intensely in WT mice, but reactivity was abrogated in *Nkx2-3*[−/−] mice (Fig. 3F), indicating that NKX2-3 is required for *St6gal1*-dependent BMAd expression in vivo.

## COUP-TFII overexpression induces *Madcam1* and *St6gal1* in gut capillaries

To evaluate the role of COUP-TFII in regulation of the mucosal addressins in vivo, we generated mice with inducible EC-specific overexpression (EC-iCOUP[OE]) or deficiency (EC-iCOUP[KO]) by crossing Cadherin 5-cre[ERT2] mice[17] to lox stop lox Rosa COUP-TFII (COUP[KO])[30] or to Nr2f2[tm2Tsa] (COUP[OE]) mice[31] (Fig. 4A). Blood ECs were isolated from PP of male and female EC-iCOUP[OE] mice 15 or 19 days after induction with tamoxifen and profiled by single-cell RNA sequencing (scRNAseq) (Fig. 4A). Blood ECs from tamoxifen-treated Cre- control and WT mice were also profiled. UMAP clustering separated the major capillary, arterial and venous EC populations (Fig. 4B) as well as capillary resident angiogenic progenitors (CRP)[12]. Venous EC comprised *Madcam1*-high HEC and *Selp*-high, *Madcam1*-low PCV (Fig. 4B and C). Lamina propria EC was characterized by enrichment in genes for transport and barrier formation.

CapEC in control cohorts lacked *Nr2f2* but expressed *Nkx2-3* (Fig. 4C). Tamoxifen treatment induced *Nr2f2* expression in the majority of EC including the CapEC populations and CRP, and led to their upregulation of both *Madcam1* and *St6gal1* (Fig. 4C). Notably, the regulatory effect was selective for the mucosal addressins: *Nr2f2* overexpression did not induce or alter expression of genes encoding the non-intestinal or non-organotypic venous markers *Vcam1*, *Ackr1*, *Ccl21a*, or *Selp* (Fig. 4C). Consistent with their high basal *Nr2f2* expression, HEV and PCV did not further upregulate the addressins in EC-iCOUP[OE] mice (Fig. 4C). Immunolabeling and confocal imaging confirmed widespread ectopic MAdCAM1 expression in capillaries in

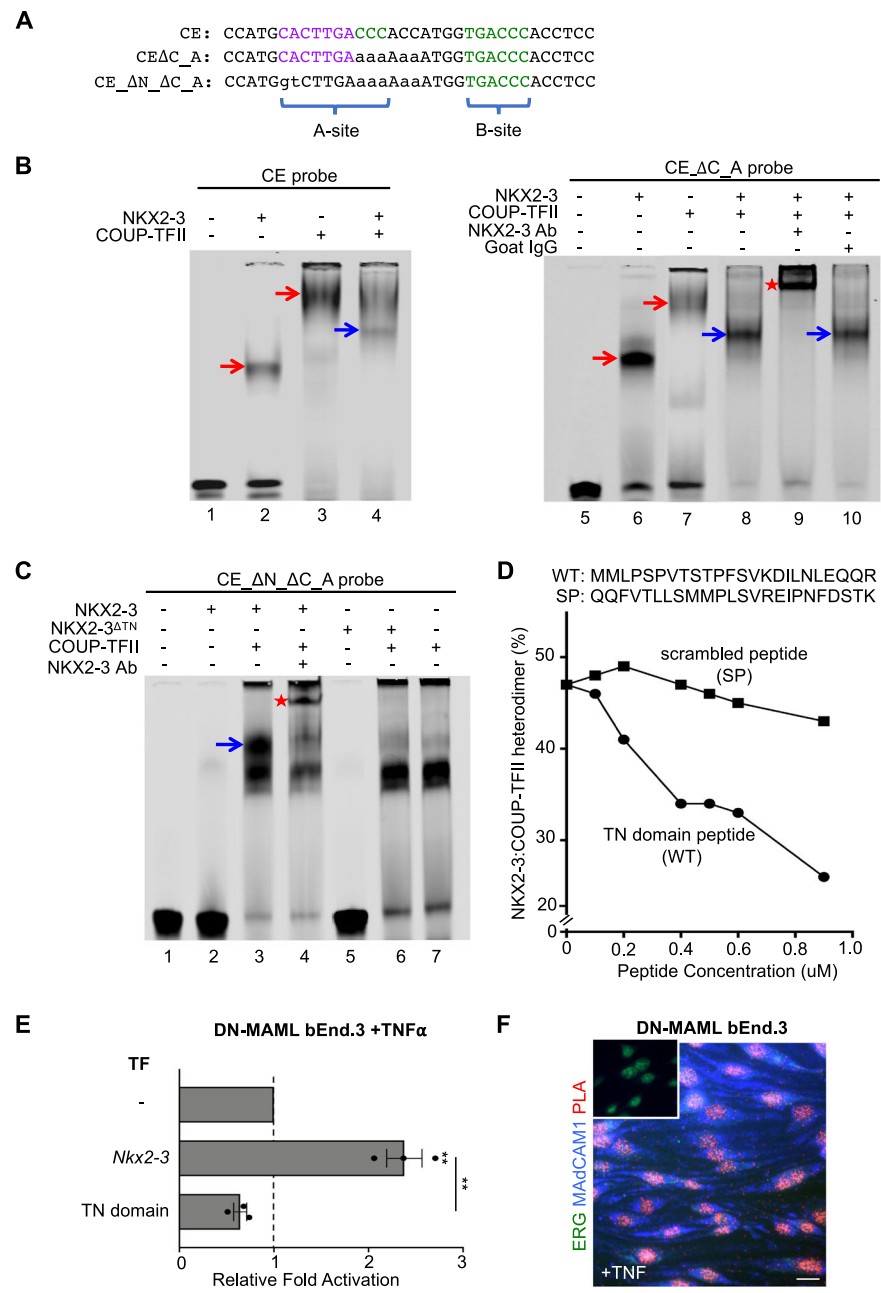

**Fig. 2 | NKX2-3 and COUP-TFII heterodimerize via the tinman (TN) domain to activate MAdCAM1. A** Sequence of wildtype and mutant CE probes for EMSA. **B** EMSA showing migration of recombinant NKX2-3 and COUP-TFII bound to the CE probe (left) and to CE_ΔC_A probe that lacks the COUP-TFII "A" site (right). Co-incubation of NKX2-3 and COUP-TFII yields a distinct band (blue arrows) reflecting an NKX2-3:COUP-TFII DNA complex that is supershifted by anti-NKX2-3 antibody but not by control antibody. **C** EMSA of protein complexes formed by incubation of the indicated TFs with a mutant CE probe lacking the NKX2-3 and the COUP-TFII "A" binding sites (CE_ΔN_ΔC_A). The probe binds COUP-TFII (lane 7) and not NKX2-3 (lane 2), but seeds a COUP-TFII:NKX2-3:DNA complex (blue arrow) when co-incubated with both TFs. The heterodimer-DNA complex is supershifted by anti-

NKX2-3 antibody (red star). NKX2-3^ΔTN failed to heterodimerize with COUP-TFII (lane 6). **D** Disruption of the NKX2-3:COUP-TFII heterodimer-DNA complex by a competing NKX2-3 TN domain peptide (WT) or by a scrambled peptide (SP). Data are expressed as percent of probe bound by NKX2-3:COUP-TFII heterodimer. **E** *Madcam1* expression in TNFα-treated DN-MAML bEnd.3 cells stably over-expressing *Nkx2-3* or the *Nkx2-3* TN domain, as evaluated by real-time PCR. Values are mean ± SEM of three biological replicates. Two tailed t-test, paired. **: $p$ value < 0.01. **F** Proximity ligation assay of NKX2-3 and COUP-TFII in TNFα-treated DN-MAML bEnd.3 cells stained for MAdCAM1 (blue) and ERG (green), with the proximity ligation signal shown in red. Inset shows a negative control with isotype-matched primary antibodies. Scale bar: 20μm.

the gut lamina propria and in PP of EC-iCOUP^OE mice (Figs. 4D and S6A). Ectopic MAdCAM1 was observed in the capillary arcades of the small and large intestinal villi as early as 14 days after tamoxifen treatment (Fig. 4D). Conversely, induced pan-endothelial COUP-TFII deficiency in EC-iCOUP^KO mice led to reduced MAdCAM1^+ HEV in PP (Fig. S6B). To assess COUP-TFII regulation of MAdCAM1 in extra-intestinal tissues, we evaluated MAdCAM1 expression in the lung and

pancreatic EC by flow cytometry. MAdCAM1 was ectopically expressed in EC-iCOUP^OE mice in pancreatic EC, which express *Nkx2-3*. In contrast, MAdCAM1 remained negative in the lung, where ECs in the adult lack *Nkx2-3* or other *Nkx2* family members[32] (Fig. S7).

SNA-binding glycotopes in vivo are less restricted to HEV than MAdCAM1. In particular, many arteries and peripheral capillary segments arising from arteries in the intestinal lamina propria also bound

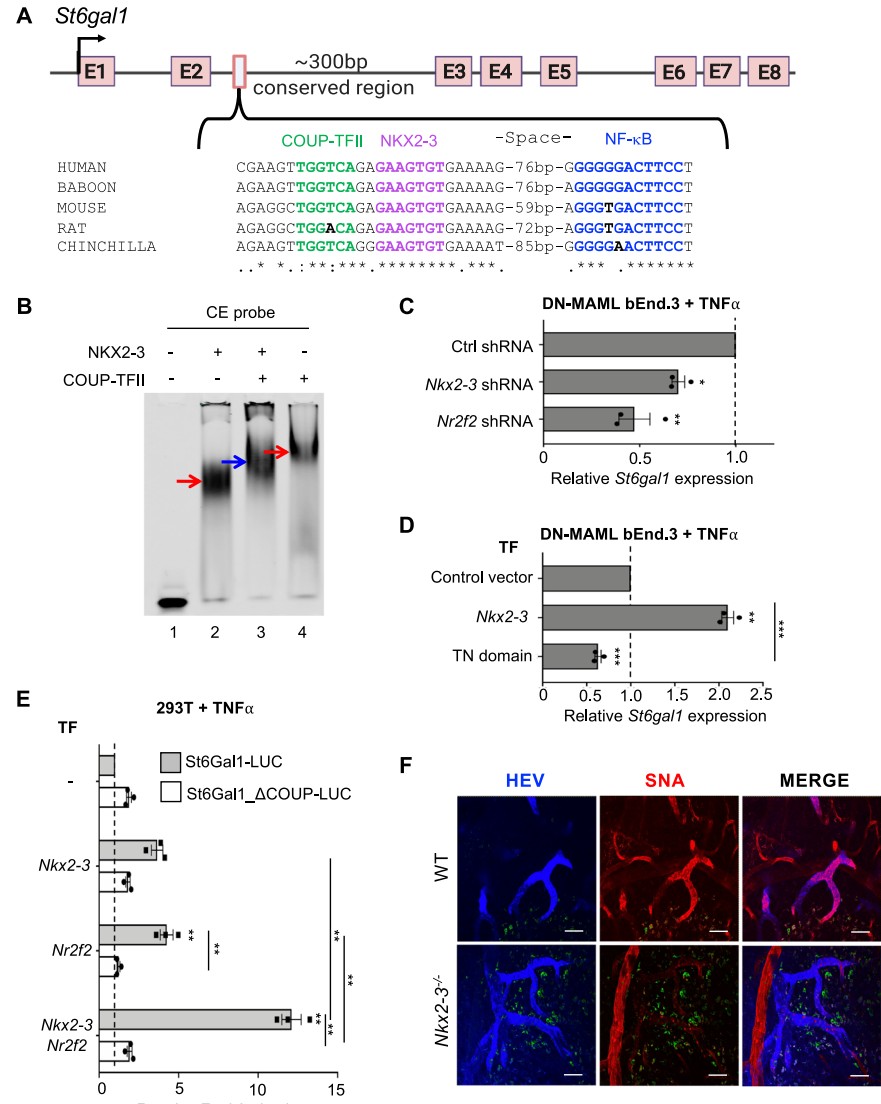

**Fig. 3 | NKX2-3 regulates *St6gal1* expression via a conserved NCCE. A** Schematic of the *St6gal1* gene and an NCCE in the second intron. **B** EMSA showing migration of NKX2-3, COUP-TFII-GST (red arrows), and NKX2-3:COUP-TFII heterodimer (blue arrow) bound to an *St6gal1* CE probe. **C** *St6gal1* expression in DN-MAML bEnd.3 cells stably transfected with *Nkx2-3* or *Nr2f2* shRNA, evaluated by real-time PCR. **D** *St6gal1* expression in DN-MAML bEnd.3 cells stably transfected with NKX2-3 or the NKX2-3 TN domain. **E** Activity of luciferase reporter driven by *St6gal1* NCCE-containing promoter (St6Gal1-LUC; gray bars) or *St6gal1* NCCE-containing promoter with mutated COUP-TFII binding sequence (St6Gal1_ΔCOUP-LUC; white bars), when co-transfected with *Nr2f2* and/or *Nkx2-3* in 293T cells. **F** Immunofluorescence of alpha-2,6-sialic acid binding lectin SNA staining in PP HEV in WT vs *Nkx2-3* deficient mice (red). HEV are marked by AF450-labeled anti-addressin antibodies MECA79, MECA89, and MECA367. Autofluorescence is shown in green. Scale bars: 50 μm. Results in C-E are shown as mean ± SEM from three independent transfections. Two tailed t-test, paired. *: $p$ value<0.05; **: $p$ value<0.01; ***: $p$ value < 0.005.

the lectin in WT mice (Figs. S5 and S8). However, in induced EC-iCOUP[OE] mice SNA stained more extensively and decorated many of the EC that ectopically expressed MAdCAM1 in the mesh-like capillary network within lamina propria villi (Fig. S8), consistent with the enhanced *St6gal1* expression in EC-iCOUP[OE] CapEC (Fig. 4C).

## NCCE in the genome

The ability of NCCE to foster cooperative COUP-TFII:NKX protein interactions to activate gene expression led us to survey the genome for conserved NCCE motifs. In particular, we hypothesized that NCCE might be enriched in association with genes that drive embryonic heart morphogenesis and pancreatic beta cell specification, developmental events which are known to involve both NKX homeodomain factors and COUP-TFII[14,33]. We screened conserved genomic regions for COUP-TFII motifs (GGTC(A/G)) with adjacent NKX HD motifs. We focused on

NCCE with a 6–16 bp spacing between motif centers, which excluded overlap of the COUP-TFII and NKX sites and encompassed the range of spacings in NCCE we show biochemically to seed the heterodimer. We identified 1954 genes nearest to these NCCE (NCCE+ genes; genomic information provided in Supplementary Data 1 and in a UCSC Genome Browser session). Since NKX and COUP-TFII motifs can regulate genes independently, as a control we identified genes lacking NCCE but containing conserved syntenic COUP-TFII and NKX HD motifs separated by larger spacings (30–60 bp), less likely to foster efficient DNA-dependent heterodimers (NCCE- genes). As additional controls, we identified genes with conserved NKX motifs 6–16 bp from scrambled motifs GTAC(G/C), AGTC(G/C), and TGGA(C/T).

Compared to control genes, NCCE+ genes are enriched in association with genes for morphogenesis of the heart and for pancreatic beta cell specification, as hypothesized (Fig. 5A),

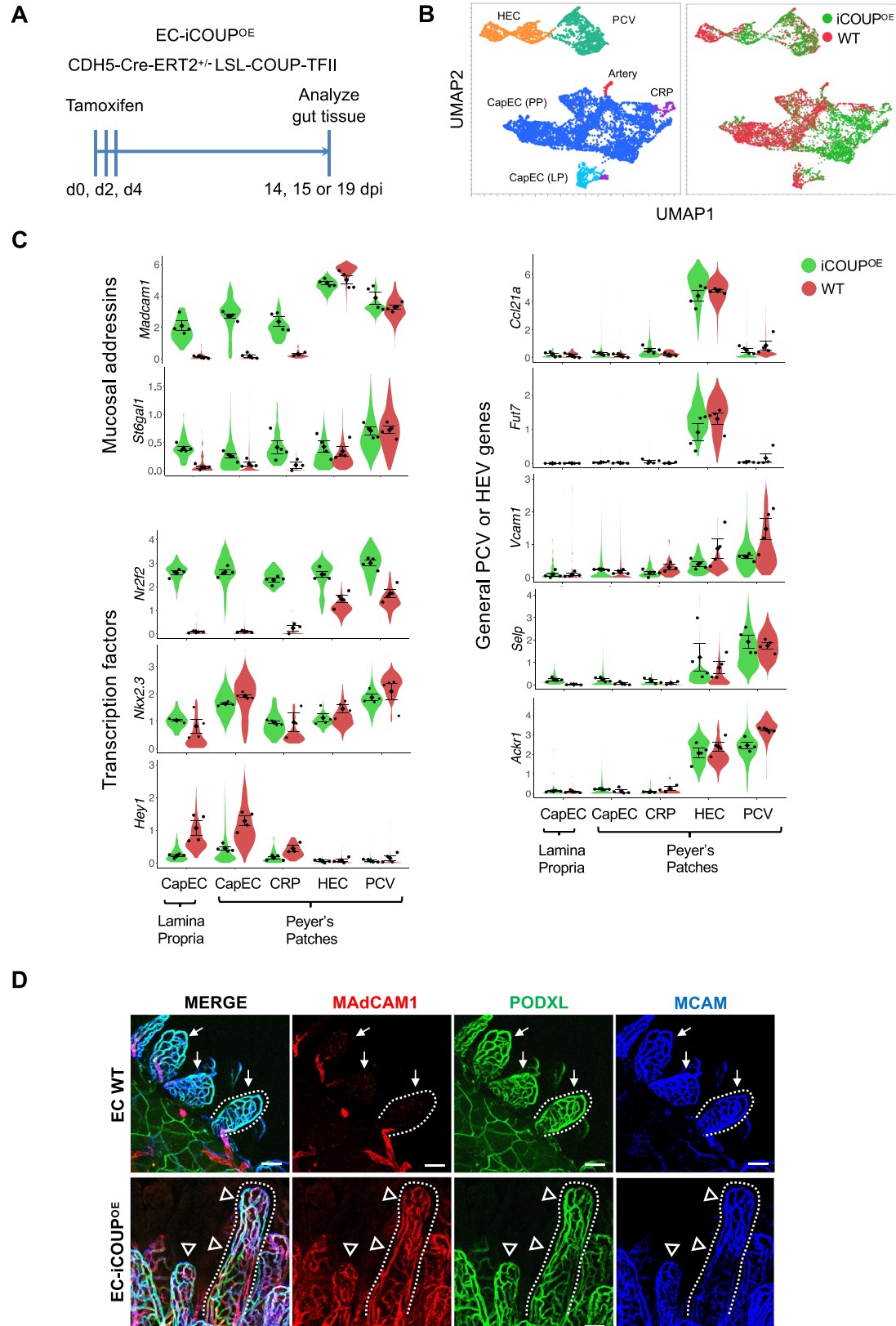

**Fig. 4 | Pan-endothelial COUP-TFII expression induces ectopic *Madcam1* and *St6gal1* in intestinal capillaries.** **A** Experimental timeline for induction of COUP-TFII in EC. **B** UMAP plot of PP BEC colored by EC subset (left) or by genotype (right). **C** Violin plots showing expression of select genes in EC-iCOUP^OE vs control BEC subsets. Dots represent mean gene expression of separate cohorts (male and female cohorts from two independent experiments), and are presented with SEM. **D** Confocal imaging showing MAdCAM1 expression in lamina propria capillaries in EC-iCOUP^OE (arrowheads) but not control (arrows) mice. Scale bars: 50 μm, EC stained by i.v. injection of directly conjugated antibodies. 20×, whole mount. See also Supplementary Fig. 6A for MAdCAM1 extension into capillary segments in PP.

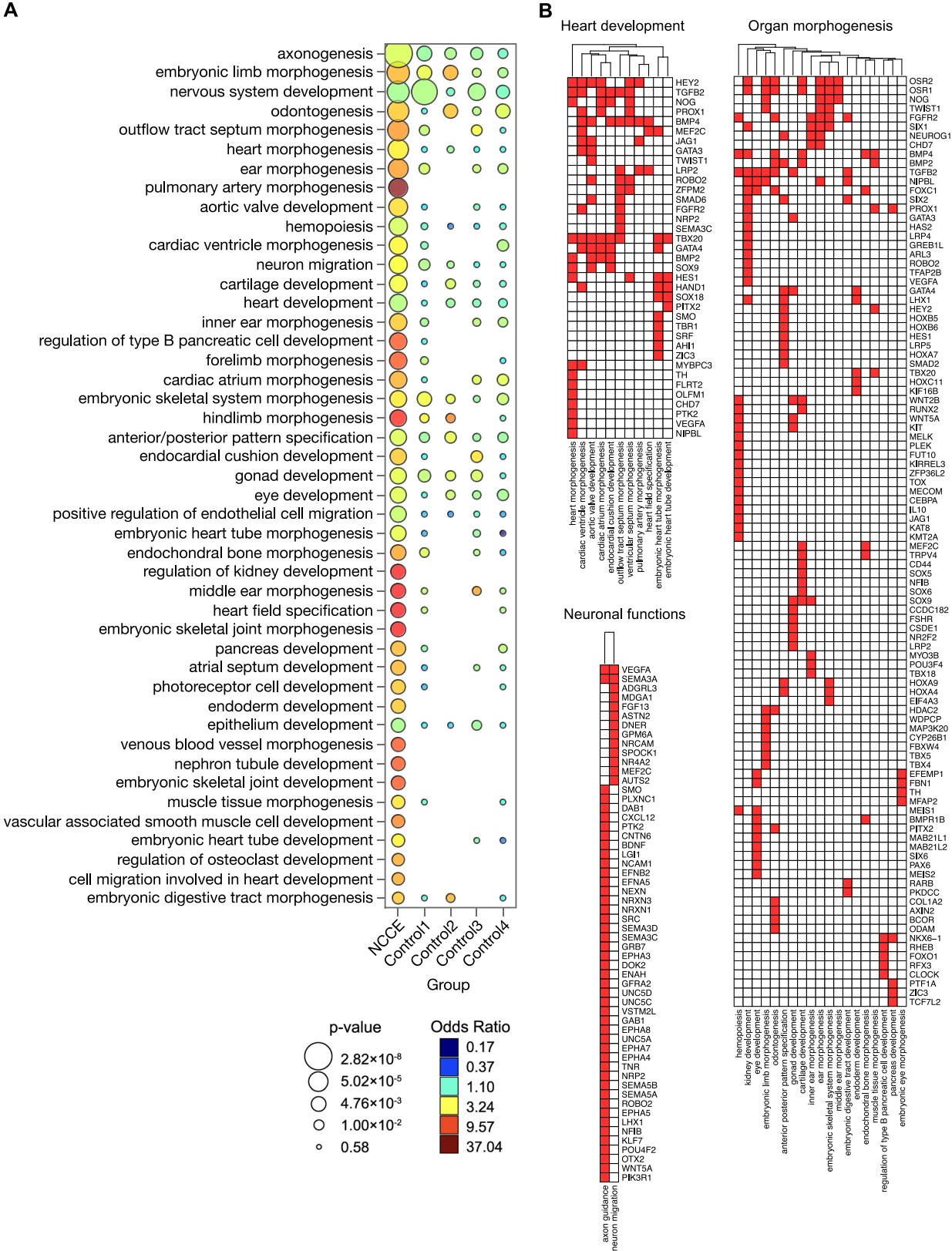

**Fig. 5 | Genome-wide NCCE are associated with organ morphogenesis.**
**A** Select GO terms enriched in NCCE+ genes compared to controls. NCCE+ genes are defined as genes nearest to conserved NCCE, where the NCCE are defined as having a 6–16 bp spacing between the centers of a COUP-TFII and an NKX HD motif. Control 1: genes associated with conserved COUP-TFII and NKX HD motifs separated by larger spacings (30–60 bp). Controls 2–4: genes associated with conserved NKX motifs 6–16 bp from conserved scrambled motifs GTAC(G/C), AGTC(G/C), or TGGA(C/T), respectively. **B** NCCE+ genes in select pathways associated with heart development, neuronal functions, and organ morphogenesis.

but also in genes involved in morphogenesis in multiple organ systems: neuron development and axonogenesis; morphogenesis of arteries, teeth, digestive tract, inner ear, cartilage, skeletal muscle, hind limb, and joints (Fig. 5B). Literature search identifies involvement or regulated expression of *Nkx* family members and *Nr2f2* in several of these processes, but their interplay has not been highlighted.

To confirm the potential of candidate NCCE to facilitate the cooperative interaction of COUP-TFII and NKX proteins, we carried out in vitro analyses of NCCE in regulatory regions of three NCCE+ genes, *Tbx5*, *Nrp2*, and *Etv2* (Fig. S9). These genes were selected for the following reasons: 1) The NKX binding motifs within them are representative of the diversity of candidate NKX motifs identified informatically, 2) They are induced developmentally in cells that co-express *Nr2f2* and *Nkx*[22,34,35] (shown also in Fig. S10), and 3) Their expression and function in embryogenesis depend on both NKX family HD proteins and on COUP-TFII. NKX2-5 with COUP-TFII regulates *Tbx5* and *Etv2* in cardiac development[15,21,36]. Moreover, the NKX motif in the *Etv2* NCCE has been previously shown to bind NKX2-5[21,37]. EMSA confirmed that the conserved *Tbx5* and *Etv2* NCCE support formation of COUP-TFII:NKX2-5 heterodimers (Fig. S11A, B and D), and the *Nrp2* NCCE seeds formation of COUP-TFII:NKX2-1 heterodimers (Fig. S11A and F). Moreover, tandem NCCE sequences from each of these genes integrated COUP-TFII and NKX2-5 or NKX2-1 to cooperatively activate reporter expression (Fig. S11C, E and G, respectively). Taken together, these results define a library of conserved genomic NCCE, confirm that diverse NCCE can seed functional NKX-COUP-TFII heterodimers, and suggest a role for NCCE in cell specification and morphogenesis in development.

## Discussion

We describe an NKX-COUP-TFII composite element (NCCE) that functions as a genomic address code for organ- and venule-specific expression of the mucosal vascular addressins MAdCAM1 and BMAd. We show that the NCCE binds developmentally programmed TFs NKX2-3 and COUP-TFII, which form heterodimers that activate transcription of the addressin genes in concert with NFκB signals. The NKX tinman domain, which is required for the interaction of NKX2-3 with COUP-TFII, is known to recruit a Groucho co-repressor complex that mediates transcriptional repression[38]. Association with COUP-TFII may mask tinman binding to Groucho or alter its protein partners to activate transcription. We show that NKX2-3 is required for *St6gal1*-dependent BMAd expression by intestinal Peyer's patch HEV, as shown previously for MAdCAM1[39]. Moreover, transgenic endothelial expression of COUP-TFII leads to ectopic expression of *Madcam1* and *St6gal1* in intestinal capillaries which express NKX2-3. Finally, we show that the *Madcam1* CE comprises an additional control element that binds capillary/artery-expressed repressors downstream of Notch signaling, reinforcing the selectivity of MAdCAM1 for venules. Together the results show that the conserved CE integrate master intestinal, venous, and capillary/arterial TFs for tight control of addressins that regulate lymphocyte homing into the intestines and their associated lymphoid tissues.

In contrast to its selective expression by intestinal venules in the adult, MAdCAM1 is widely displayed by endothelial cells including PLN HEV in early neonatal life[40]: it is downregulated in PLN with concurrent upregulation of PNAd by 4–6 weeks of age. NKX2-3 deficiency has no effect on perinatal MAdCAM1. Instead, intestinal HEV[41] as well as splenic marginal zone EC[42] become PLN-like in adults, expressing PNAd. Our finding that NKX2-3 deficiency abrogates PP HEV BMAd expression is consistent with this "transdifferentiation" towards a peripheral HEV phenotype. Mechanisms underlying the NKX2-3-independent *Madcam1*

expression in neonates and the NKX2-3-dependent suppression of PNAd in adult gut HEV remain to be defined.

Genome-wide association studies implicate NKX2-3 variants in susceptibility to both Crohn's disease and ulcerative colitis[43], diseases in which MAdCAM1-dependent lymphocyte recruitment is implicated and upregulated[9,44–46]. While the basis for these associations remains unclear, allelic studies show that *NKX2-3* risk haplotypes are over-expressed compared to non-risk alleles in the colon[44]; and we show that *NKX2-3* cell instrinsically enhances endothelial expression of MAdCAM1. These results could support a proposed link between variant *NKX2-3* and endothelial cell regulation of immune cell traffic and adhesion.

The other key component of the NCCE control complex, COUP-TFII, is an evolutionarily conserved orphan nuclear receptor that regulates diverse biological processes including angiogenesis, cardiac chamber development, neuronal subtype specification, circadian rhythm, and metabolism[14,16,47]. During embryonic endothelial fate specification, COUP-TFII supports venous differentiation by repressing Notch signaling and arterial differentiation[18], and its expression is maintained in most venular endothelial cells in situ in the adult[12,13]. COUP-TFII forms stable homodimers in solution that bind to direct repeats (DR) of the COUP-TFII half-site motif with variable spacing between repeats. Depending on the spacing, these DR sequences can also bind heterodimers of COUP-TFII with the nuclear hormone receptor RXR[48,49]. COUP-TFII is classically repressive in these settings[49,50]. COUP-TFII can also regulate gene expression positively in complex with other DNA-binding TFs (e.g., SP1, HNF4, or PROX1) independently of cognate COUP-TFII DNA binding sites[51–55]: These regulatory complexes are not associated with conserved genomic sites for COUP-TFII-DNA interaction, and thus appear to be driven by protein-protein recognition rather than by DNA codes. In contrast, while our gel shift assays show that DNA-bound COUP-TFII can recruit NKX2-3 in the absence of a canonical NKX motif, this interaction is not sufficient to drive reporter expression in our systems: Activating NKX-COUP-TFII heterodimers may require stabilization by canonical NCCE in which both TFs bind their cognate motifs.

COUP-TFII and NKX family TFs coordinate to regulate cell fate decisions in development. For example, NKX2-5, the vertebrate homolog of Drosophila "tinman", establishes the embryonic heart field[20] and COUP-TFII drives atrial fate determination within the heart field by inducing TBX5, a master transcription factor for atrial development[15]. COUP-TFII and NKX2-5 also control pro-epicardial and endocardial development, mediated in part by induction of *Etv2*[15,21]. COUP-TFII and NKX2-1 coordinately regulate expression of *Nrp2* in E13.5 GABAergic neurons, triggering their migration toward the amygdala or neocortex[22–24]. We find conserved NCCE in *Tbx5*, *Etv2*, and *Nrp2* that support NKX:COUP-TFII dimerization and cooperative transcriptional activation in reporter assays. NCCE are also enriched among genes implicated in pancreatic beta cell fate specification (regulated by COUP-TFII and NKX2-2[56–58]); in odontogenesis (which involves COUP-TFII and NKX2-3[59,60]); in blood developmental hemopoeisis; and in morphogenesis or development of ear, kidney, muscle, cartilage, and limbs. In contrast, genes with NKX and COUP-TFII sites that are well separated (>30 bp) show significantly less enrichment in these GO terms for morphogenesis or organogenesis. While the physiologic roles of the individual NCCE presented here as a resource remain to be determined, taken together the results suggest that NCCE may contribute to diverse cell fate and morphogenetic events in vertebrate organogenesis. Our findings highlight the adoption of fundamental morphogenetic mechanisms for control of tissue specific vascular and immune organ specialization.

## Methods

All experiments were approved by the accredited Department of Laboratory Animal Medicine and the Administrative Panel on Laboratory Animal Care at Stanford and the VA Palo Alto Health Care System.

### Plasmid constructs

mCE-NF$\kappa$B and mNF$\kappa$B ($-422$ and $-260$ bp upstream of the *Madcam1* transcription start site, respectively) were PCR-amplified as a *Kpn*I-*Hin*dIII fragment from mouse genomic cDNA and cloned into pGL4.11[*luc2P*] to generate the luciferase (LUC) reporter constructs. To analyze the NKX2-3 and NR2F2 binding sites at CE, site-specific mutations were introduced into CE_$\Delta$N-NF$\kappa$B-LUC via dual PCR and for CE_$\Delta$C_A, and CE_$\Delta$C_AB via Cyagen custom service. CE-LUC, used for control in mutational studies, and human *MADCAM1* reporter constructs hCE-LUC and hCE_NF$\kappa$B-LUC were made by Cyagen custom service. Control Renilla (Ren) Luciferase vector is from Promega. For co-transfection studies, mammalian expression vectors *Nkx2-3*, *Nr2f2* (COUP-TFII), and *Hey1* were obtained from ABM custom service (Richmond, BC, Canada). All plasmids are described in Supplementary Table S3.

*Nr2f2*, *Hey1*, and *Nkx2-5* cDNA were PCR-amplified as *Bam*HI-*Eco*RI-fragments and *Nkx2-3* and *Nkx2-3Δ2-24* cDNA were PCR-amplified as *Eco*RI-*Hin*dIII- fragments from mouse cDNA and cloned into the pET21-A vector (Novagen) to generate His and T7-tagged recombinant proteins. *Nr2f2* cDNA was cloned into the pGEX2TK vector (GE Healthcare) to generate GST-tagged COUP-TFII. pET21A-*Nkx2-1* was from Cyagen, a custom service. The fidelity of all constructs was verified by sequencing. Primer sequences are provided in Supplementary Table S1.

### Cell lines

bEnd.3 and HEK293T cells were obtained from ATCC (CRL-2299 and CRL-3216 respectively). HEK293T cells were grown in Dulbecco's modified Eagle's medium containing high glucose and L-pyruvate and supplemented with 10% heat-inactivated fetal bovine serum at 37 °C and 5% $CO_2$. bEnd.3 and DN-MAML bEnd.3 cells were grown in the same media supplemented with penicillin-streptomycin. BEnd.3 and bEnd.3-based transductions were grown and maintained at confluence until the cells assumed an organized appearance (typically 2–3 days after confluence) prior to stimulation with TNFα.

### Stable transfections

The bEnd.3-DNMAML cell line was constructed by transducing bEnd.3 cells (ATCC CRL-2299) with MSCV-MAML(12-74)-EGFP, which contains amino acids 12–74 of human MAML fused in frame with EGFP (Gift from Warren Pear, University of Pennsylvania). EGFP-positive cells were sorted on a BD FACS Aria III. Retrovirus was produced with the Phoenix-Eco packaging cell line (ATCC CRL−3214) using standard methods[61].

Stable bEnd.3 DN-MAML cell lines overexpressing *Nkx2-3*, *Nkx2-3* TN domain or control vector, and bEnd.3 cell lines expressing *Hey1*, *Nkx2-3*, *Nr2f2*, as well as shRNA for *Nkx2-3*, *Nr2f2*, and control shRNA, were generated by transduction with respective retroviral constructs packaged in HEK293T cells via standard protocols (Invitrogen).

### Transient transfection assays

For luciferase reporter assays, $1.5 \times 10^5$ HEK293T cells were plated on 24-well plates followed by transfection after reaching ~70–80% confluency. The CE_NF$\kappa$B-LUC and NF$\kappa$B-LUC constructs were amplified from a pGL4 vector (Promega). CE_$\Delta$N-NF$\kappa$B-LUC, CE_$\Delta$C_A, CE_$\Delta$C_AB, mouse CE-LUC, and human CE-LUC were amplified from a pGL4.23-based vector custom-made by Cyagen (LUC reporter containing a minimal promoter). Renilla (Ren) Luciferase was amplified from a pRL vector (Promega). 36 h after TNFα stimulation, cells were harvested by centrifugation and re-suspended in lysis buffer. For Fig. 1H, cells were

stimulated for 16 h. Luciferase assays were carried out using the Dual-Glo Luciferase assay system (Promega) and a Turner, TD-20/20 luminometer as previously described[62]. Firefly luciferase values were normalized to Renilla luciferase signal of the TNF-stimulated conditions from at least three independent experiments each with 2−3 replicates. Statistical significance was assessed via Student's one-tail or two-tail paired t-test.

### Protein expression and purification

The pET21A-*Hey1*, -*Nr2f2*, -*Nkx2-3*, -*Nkx2-3Δ2-24*, -*Nkx2-1*, -*Nkx2-5*, and pGEX2TK-*Nr2f2* protein expression plasmids were transformed into *E. coli* BL21 competent cells. Protein expression and purification were carried out as previously described[63] and purified proteins were quantified against BSA.

### Electrophoretic mobility shift assay (EMSA)

EMSAs were performed as previously described[63]. Probes were generated by annealing 100 pmol of sense and antisense oligonucleotides (Table S1) and 1–2 pmol of probe was used in each reaction. Gel shift reactions were conducted at 4 °C in 20% glycerol, 20 mM Tris (pH 8.0), 10 mM KCl, 1 mM DTT, 12.5 ng poly dI/C, 6.25 pmol of random, single-stranded oligonucleotides, BSA and the probe in the amounts specified above. All samples involving only NKX2-3 or HEY1 were loaded on an 8% gel, whereas any involving COUP-TFII were loaded on a 6% gel to resolve protein−DNA complexes. In reactions with cold competitors, 20x unlabeled probes were included in the reactions. Antibodies against specific proteins (anti-NKX2-3, anti-HRT1, Nkx2-1 and Nkx2-5 were from Santa Cruz Biotechnology: sc-83438x, sc-16424x, SC-53136x, and sc-376565X, respectively); anti-COUP-TFII from Perseus Proteomics (PP-H7147-00); and anti-T7 antibody from Abcam (ab97964) (Table S2). Antibodies were added to reactions at the same volume amount of the respective protein to obtain super-shifts. All gels were imaged using the Odyssey Imaging System from Licor.

For Fig. 2D, NKX2-3 and COUP-TFII were incubated with the indicated NCCE probe, in the presence or absence of the indicated concentrations of the TN domain peptide or scrambled peptide. Results are presented as percentage of total probe within the formed heterodimers, calculated by quantifying the amount of NKX2-3:COUP-TFII complex (using the Odyssey software), and dividing this quantity by the total amount of probe present (calculated as the sum of signals from the NKX2-3:COUP-TFII complex and from the free probe).

### Chromatin immunoprecipitation (ChIP)

ChIP was performed using the ChIP-IT High Sensitivity Kit (Active Motif), per manufacturer's instructions. Cross-linked chromatin isolated from DN-MAML bEnd.3 cells was immunoprecipitated using antibodies against COUP-TFII (Table S2) and the corresponding isotype control. We used PCR primers (Table S1) spanning CE along with a genomic control region (not containing a COUP-TFII binding site). The input and ChIP samples were subjected to real-time PCR (Biorad) and analyzed to ensure that amplification was in the linear range. Three biological samples along with three technical replicates were performed. The data were analyzed as previously described[63]. Error bars indicate the standard error of mean. Statistical significance was assessed via two-tailed T-test, paired.

### Proximity ligation assay (PLA)

PLA (Sigma) was used to identify specific co-localization of NKX2-3 and COUP-TFII in fixed cells, following manufacturer's protocols with a few modifications. Prior to the PLA, cells were fixed in acetone at 4 °C for 10 min. Cells were incubated with anti-NKX2-3 (Atlas Antibodies, Table S2) and anti-COUP-TFII (Perseus Proteomics; Table S2) or normal rabbit IgG or mouse IgG2a (Santa Cruz Biotechnology) controls. Following PLA, slides were blocked with 10% normal rabbit serum for 30 min at RT and incubated with anti-ERG-488 (Abcam) overnight at

4 °C. The slides were then incubated with antibody against Dylight405-conjugated anti-MAdCAM1 (MECA367) for 2 h at room temperature and imaged.

## RNA isolation, real-time RT-PCR, and microarray analysis

Total RNA was extracted using TRIzol followed by the RNeasy mini kit (Qiagen) cleanup and RQ1 RNase-free DNase set treatment (Promega) according to the manufacturer's instructions. First-strand cDNA was synthesized Superscript II (Invitrogen). TaqMan universal master mix reagents (Applied Biosytems) were used for Realtime RT-PCR assays. The primers/probes used in this study are listed in Supplementary Table S1.

Microarray analysis of *Nkx2-3*, *Nr2f2*, and *Hey1* in bEnd.3 cells were performed as previously described[10] using Affymetrix Genechip Mouse Gene 1.0 ST arrays[10]. Raw expression values above 120 are considered positive.

## Animals

COUP-TFII[OE/OE31] or COUP-TFII[KO/KO30] mice were crossed with CDH5(PAC)Cre[ERT2] mice[17] to generate tamoxifen-inducible endothelial cell-specific COUP-TFII OE or KO mice (EC iCOUP[OE or KO]). Littermates not harboring the cre transgene were used as controls. 200 μg/g of tamoxifen was injected three times, i.p. every other day. For the SNA and COUP-TFII[KO/KO] studies, 100 μg/g of tamoxifen was i.p. injected every other day three times. Days post-injection correspond to the number of days after the first injection. *St6gal1*[-/-] mice[64] were from the Marth lab. *Nkx2-3*[-/-] mice[19] were backcrossed to BALB/cJ mice and maintained at the Department of Immunology and Biotechnology, University of Pécs[42]. C57BL/6J mice were obtained from The Jackson Laboratory, Bar Harbor, USA. All animals were housed under standard conditions, maintained in a 12 h/12 h light/dark cycle at $22 \pm 2\,°C$ and given food and tap water ad libitum. Unless otherwise stated, male and female mice between the ages of 6–10 weeks were used in all experiments.

## Single-cell sequencing

Endothelial cells from the Peyer's patches of EC-iCOUP[OE] and control mice were isolated 14–19 days post tamoxifen treatment, dissociated, and sorted as previously described[12]. Single-cell gene expression was assayed using the 10x Chromium v3 platform using Chromium Single Cell 3' Library and Gel Bead Kit v2 (10X Genomics, PN-120237) according to 10X Genomics guidelines. Male and female cohorts were processed together and resolved post-sequencing. Libraries were sequenced on an Illumnia NextSeq 500 using 150 cycles high output V2 kit (Read 1: 26, Read2: 98, and Index 1:8 bases). The Cell Ranger package (v3.0.2) was used to align high-quality reads to the mm10 transcriptome (quality control reports available: https://stanford.io/37sXZV3). Quality control and data analysis were carried out as described[12]. Briefly, normalized log expression values were calculated using the scran package[65]. Imputed expression values were calculated using a customized implementation (https://github.com/kbrulois/magicBatch) of the MAGIC (Markov Affinity-based Graph Imputation of Cells) algorithm[66] and optimized parameters ($t = 2$, $k = 9$, ka = 3). Supervised cell selection was used to remove cells with non-blood endothelial cell gene signatures: lymphatic endothelial cells (Prox1, Lyve1, Pdpn); Pericytes (Itga7, Pdgfrb); fibroblastic reticular cells (Pdpn, Ccl19, Pdgfra); lymphocytes (Ptprc, Cd52). The Arterial (*Gkn*+), HEC, PCV, and CRP clusters were defined based on canonical marker expression. Batch effects from technical replicates were removed using the MNN algorithm[67] as implemented in the batchelor package's (v1.0.1) fastMNN function. Dimensionality reduction was performed using the UMAP algorithm. For UMAP embeddings, cell-cycle effects were removed by splitting the data into dividing and resting cells and using the fastMNN function to align the dividing cells with their resting counterparts. Violin plots were generated using ggplot2; *y*-axis units

for gene expression data corresponding to log-transformed normalized counts after imputation. Mean gene expression for male and female cells from each subset was calculated for each sample and plotted with standard errors. With the exception of mature *Gkn3*[+] arterial EC (identifiable among captured cells only in the WT cohort), and of CRP (missing in the male WT mice), all subsets were represented by cells within both male and female mice from each condition.

## Identification of conserved NCCE

Conserved genomic regions were defined as regulatory elements with log-odds score greater than 300 in UCSC Genome Browser's phastCons Placental Elements track, which aligns 60 vertebrate species and measures evolutionary conservation of genomic sequences[68,69]. COUP-TFII and NKX motifs were searched within each conserved region in the mouse genome using HOMER[70]. The spacing between the two motifs within a NCCE was calculated as the distance between the centers of the motifs. Genome coordinates of the motifs were converted from mouse (mm10) to human (hg38) using the liftOver tool[71], and only mouse coordinates whose syntenic human counterparts correspond to COUP-TFII and NKX binding motifs were defined as conserved NCCE. All NCCEs are provided as a custom track in a UCSC Genome Browser session: https://genome.ucsc.edu/s/m.xiang/NCCE_gap6%2D16_conserved.

## Imaging

PP were imaged following either retro-orbital injection of fluorescent-labeled antibodies or immunofluorescence staining of LN sections. Injected antibodies (25–75 μg) were administered 5–30 min prior to sacrifice and PP removal. To image the overall vasculature, the PP was compressed to ~35–50 μm thickness by applying gentle pressure on a coverslip on a glass slide (Figs. 4D, S5, and S6). Alternatively, tissues were fixed with 4% paraformaldehyde (PFA), cryoprotected with sucrose, frozen in OCT (Sakura® Finetek) in 2-methylbutane (Sigma) on dry ice and stored at −20 °C. 50 μm cryo-sections were stained with antibodies as previously described[12]. Cell lines were fixed with 4% PFA and stained according to standard protocols for imaging.

For *Sambucus nigra* lectin (SNA) studies, binding to the vascular endothelium was assessed by perfusing mice with fluorescently labeled lectin. Briefly, mice were injected iv with a cocktail of fluorescent antibodies to mark endothelial cells and HEV (6 μg anti-CD146, 25 μg anti-mouse MAdCAM-1 (clone # MECA367), 25 μg anti-mouse MAdCAM-1 (clone # MECA89) and 36 μg anti-mouse PNAd (clone # MECA79) 15 min prior to anesthesia; followed 5 min later by i.p. injection of 5 units of Heparin (Sigma-Aldrich, H3149). Mice were anesthetized by isoflurane and subjected to vascular perfusion by gravity (75 cm in height) flow with $Ca^{2+}Mg^{2+}$ HBSS buffer containing 3 μg/ml SNA (Vector Laboratories). A flow rate of ~2 ml/min was maintained. After perfusion with 20 ml SNA solution, mice were sacrificed, and tissues were positioned in Fluoromount-G (Southern Biotech) and manually compressed by firm pressure on an overlying coverslip for confocal imaging. The slides were imaged using Apotome 2.0 fluorescence microscope or LSM 880 laser scanning microscope (Zeiss).

For the *Nkx2-3*[-/-] SNA studies, BALB/c mice were co-housed with *Nkx2-3*[-/-] mice under standard conditions and were used between 6 and 8 weeks of age. Mice were injected with cocktails containing APC-conjugated anti-CD31 (clone # MEC13.1, BD Biosciences), anti-mouse PNAd (clone # MECA-79), and anti-mouse MAdCAM-1 (clone # MECA-89) labeled with DyLight405 in PBS at 200 μl final volume. 20 min later the mice were anesthetized with ketamine-xylazine injection i.p. supplemented with 0.5 mg/kg atropine. The anesthetized mice were injected with heparin, followed by sequential perfusion through the left ventricle with 10 ml PBS followed by PBS with 2 μg/ml *Sambucus nigra* lectin (SNA) conjugated with Cy3 (Vector Laboratories) containing 0.1 mM $CaCl_2$ of the period of 20 min. After washing with PBS the animals were perfusion-fixed by 4% buffered paraformaldehyde.

After fixation, the inguinal lymph nodes and Peyer's patches were placed onto glass histological slides in an area demarcated with PAP-pen to prevent the spillage of mounting medium. Two strips of double-sided sticky tape were placed outside the marked area, and 50 μl of 1:1 mixture of PBS-glycerol was added, in which the samples were immersed, and covered with 22 mm × 40 mm glass coverslips. The samples were viewed using an Olympus Fluoview FV-1000 laser scanning confocal imaging system (Olympus Europa SE & Co. KG, Hamburg, Germany).

For NKX2-5 and COUP-TFII co-localization studies (Fig. S10), BALB/c embryos at E10.0 were isolated in PBS and then fixed in 4% PFA overnight at 4 °C. The following day, embryos were washed in PBT (PBS containing 0.1% Tween-20), dehydrated in an ascending methanol sequence, xylene treated, and embedded in paraffin. Immunofluorescence was performed on 7 μm deparaffinized sections. Briefly, sections were subjected to antigen retrieval in Tris buffer pH 10.2 for 8 min, washed in 0.1% PBT and incubated in blocking buffer (0.2% milk powder, 99.8% PBT) for 1 h at room temperature. Primary antibodies were incubated in blocking buffer overnight at 4 °C. The following day, the sections were washed three times with PBT and incubated for 2 h with corresponding secondary antibodies in blocking buffer at room temperature. After three washes in PBS, DAPI (Sigma-Aldrich) was added to counterstain the nuclei. The sections were mounted using Prolong Gold Antifade Reagent (Invitrogen) and imaged using Zeiss LSM-700 confocal microscope. The following primary antibodies were used: ERG (Abcam), NKX2-5 (Santa Cruz), and COUP-TFII (Perseus Proteomics). Secondary antibodies were Alexa Fluor conjugates 488, 555, and 647 (Life Technologies).

## Statistics and reproducibility

All EMSA and imaging were performed a minimum of three times. For mouse experiments, male and female of similar ages were randomly allocated to each experimental group. Statistical test used for each experiment is indicated in the figure legend.

## Reporting summary

Further information on research design is available in the Nature Portfolio Reporting Summary linked to this article.

## Data availability

The authors declare that all data supporting the findings of this study are available within the article and its supplementary information files. scRNAseq raw data are available in the NCBI Gene Expression Omnibus (GEO) repository with accession number GSE217886. Source data are provided with this paper.

## Code availability

Custom scripts for identifying conserved NCCEs are available at https://github.com/mxiang1/NCCE.

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

## Acknowledgements

We thank Milladur Rahman, Agata Szade, Sofia Nordling, Mike Lee, Michael Bscheider, Borja Moreno, Romain Ballet, Martin Brennan, Jeffrey Ye, Nicole Lazarus, Jean Jang, and Dhananjay Wagh for assistance with technical support, cell preparation, and single-cell sequencing experiments. We thank Ralf and Susanne Adams for the Ve-Cad5-creERT2 mice and J. Paulson from The Scripps Research Institute for *St6gal*1$^{-/-}$ mice. This work was supported by NIH grants R01 AI130471 and CA228019, and award I01 BX-002919 from the Department of Veterans Affairs to E.C.B. T.T.D. was the recipient of a fellowship under NIH training grant T32 AI07290 from the NIAID and T32 HL098049 from the National Heart, Lung, and Blood Institute, and was supported by an AHA Postdoctoral Fellowship and the Stanford Cardiovascular Institute. M.X. was supported by the Tobacco-Related Disease Research Program of University of California (T31FT1867). Y.W. was supported by Knut and Alice Wallenberg Foundation KAW 2018.0423. Y.Z. was supported by a CRI Irvington Postdoctoral Fellowship. K.B. was supported by NIH F32 CA200103. K.R.H. is supported by the NIH/NHLBL (R01-HL128503) and is an HHMI Investigator. S.R. and K.R.H. were funded by The New York Stem Cell Foundation (NYSCF-Robertson Investigator to KRH). P.B. is supported by OTKA grant No. 128322, GINOP 2.3.2-15-2016-00050 and EFOP 3.6.1-16-2016-00004 grants. B.G. was supported by the National Research, Development and Innovation Fund of Hungary (TKP2021-EGA-16).

## Author contributions

T.T.D. conducted or oversaw most experiments. M.X. performed genomic analyses. A.R., J.P., Y.W., N.S., Y.Z., W.R., S.R., E.O., H.K., T.M.D., Y.B., D.G., and E.P.B. performed experiments. P.B., F.G., B.G., and G.B. produced and analyzed the *NKX2-3*$^{-/-}$ mice. T.T.D., A.R., J.P. analyzed experiments. K.B. and M.X. analyzed single-cell RNA-seq data. K.R.H. provided input and advice for experiments. T.T.D., M.X., A.R., J.P., and E.C.B. wrote and edited the manuscript. J.P. and E.C.B. conceived and supervised the study.

## Competing interests

The authors declare no competing interests.
