## [Peer Review File · Nature Communications]

An NKX-COUP-TFII morphogenetic code directs mucosal endothelial addressin expressionREVIEWER COMMENTS

Reviewer #1 (Remarks to the Author):

In this manuscript Dinh et al. address the regulation of mucosal addressin expression. They show that a composite element comprising binding sites for NKX2-3 and COUP-TFII is present in regulatory regions of Madcam1 and St6gal1 and functionally relevant in vitro as well as in vivo in the context of pan-endothelial overexpression in mice. Similar NCCE were found in genes regulating morpho- and organogenesis. Thus, they conclude that these NCCE are a conserved morphogenetic code directing tissue-specific vascular and immune organ specialization.

One has to congratulate the authors on this work. The study is well conducted, uses state-of-the-art methodology and the manuscript is well written. It reports significant findings with important novelty for the field. The conclusions are accurately supported by the experiments.

I have only a few comments that might help to further improve the manuscript:

- I suggest using a color code in the bar graphs to make it easier for the reader to understand the different interventions of first view
- Fig. S2A: There seems to be a dose-dependent effect of NR2F2 on human MADCAM1 expression. Can the authors further comment on that? How can that be explained? Is this similar in mice?
- Fig. 4 and associated supplemental data: Very nice data demonstrating the ectopic expression of Madcam1 in the intestine. However, it would be important to also show data from extraintestinal tissues.
- The transition to the genome-wide screening for NCCE is quite sudden and could be better explained.
- Some typos: e.g. l. 126, l. 489, l. 493
- citation 2 is not indicated in the reference list

Reviewer #2 (Remarks to the Author):

In this manuscript, Dinh et al examine the transcriptional code involved in the venous expression of addressin molecules specifically in gut associated lymphoid tissue. Understanding the transcriptional circuitry which restricts the spatio-temporal expression of certain defining markers and in turn determines the fate of specific cell types is necessary to understand how tissue specificity arises, not only in during development, but also how it is maintained in post-natal life. As such, this work addresses the HEV tissue-specific expression of Madcam1 and St6gal1, two genes that encode addressin molecules which are important in mediating immune responses in gut associated lymphoid tissue. Using transactivation assays, the authors show that a conserved composite Nkx-CoupTF element (NCCE), found in the promoter and intron of Madcam1 and St6gal1 respectively, is required (in vitro) for directing reporter expression in 293T or b.END cells. Further using CoupTFII overexpressing mice, the authors show increased Madcam1 and St6gal1 expression in certain endothelial cell types isolated from peyer's patches of adult mice. Finally, the authors suggest that similar conserved NCCEs are found through the genome and are involved in regulating the tissue specific expression of several developmental genes.

This is an interesting paper and furthers our understanding of the tissue specific expression of addressins in GALT. Overall, I am supportive of it's publication; however I do have some concerns which if addressed would improve the quality of the paper and make it more accessible to a general audience as well. I have divided my comments based on major figures. They are as follows:

Figure 1:

1) The authors do not provide any details as to how the initial NCCE element in the Madcam1 promoter was identified – was it through specific computational analysis? Additionally, the methods section mentions ATAC-seq; however, I could not come across these data in the results section

anywhere?

- 2) The authors should explain the results of their EMSAs better in Figure S1 and not just allude cursorily to them without stating the actual result in text.
- 3) Is the composite element used in the reporter assays in Figure 1B and C exactly the same as the one in the mouse Madcam1 promoter?
- 4) In Fig. 1C, 293T+TNF α should read as 293+TNF α
- 5) How many tandem NCCE sites were used in the reporter used in experiment for Fig. 1D? Did this reporter also require additional stimulation with TNF α - a schematic would be helpful here.
- 6) The authors mention that bEnd.3 cells express low levels of Nkx2-3, CoupTFII, and Hey1. This claim should be substantiated with either a reference or data.
- 7) Statistical analysis for the data shown in Fig. 1G are missing.
- 8) It might be helpful to explain Fig. 1H in a little more detail in the text. Was TNF α also used as a stimulus in this experiment? Additionally, there is an abrogation of reporter activity in WT bEnd.3 cells which is relieved upon blocking of Hey1 in DN-MAML bEND.3 cells - does this imply that even the low level of endogenous Hey1 present in WT bEnd cells is sufficient to block activation by endogenous Nkx2-3/CoupTFII?
- 9) The p values for the graphs in Fig. S2A are missing from the legend.
- 10) In Fig. S2A, why do higher levels of Nr2f2 inhibit reporter activation in presence of Nkx2-3?

Figure 2:

- 1) Why is the Nkx2-3-COUPTFII heterodimeric gel shift band lower than that observed for binding by COUP-TFII alone \diamond Is this a result of the different sizes of the hetero/homodimeric complexes formed?
- 2) The authors should explain the peptide competition assay better in the text of the results section. Additionally, the methods section is missing information on this experiment. The Y-axis in Fig. 2D should start at 0 and not at 20 (can show a broken axis if the differences are to be made more apparent)
- 3) Why were the experiments in Fig. 2E and 2F done specifically only in DN-MAML bEND.3 cells and not in WT bEND.3 cells since Madcam1 expression can also be stimulated by TNF α and Nkx2-3 in WT cells?

Figure 3:

- 1) The NCCE in the St6gal1 locus appears to be in 2nd Intron according to the schematic depicted in Fig. 3A and not in the first intron as mentioned in the text.
- 2) Again, it would be helpful to explain the logic behind performing experiments only in DN-MAML bEnd.3 cells and not WT bEnd. 3 cells (as in Figs. 3C and D).

Figure 4:

- 1) What is the age of mice used for the scRNA-seq experiments in Fig. 4? Additionally, what is the WT genotype used for analysis? Are they pure WT mice? Tamoxifen only controls? Ideal controls would be Tamoxifen-treated Cdh5CreER tg/0 mice without the overexpressing or KO COUPTF allele since Tamoxifen+Cre induction itself is known to cause molecular changes. Either way, the authors should clearly mention what WT means in this context?

2) It appears in the tSNE plot that there is almost a complete separation of the WT and COUPTFII-OE cells in certain clusters (HEC and CapEC) – it would be interesting to note what are the gene changes driving this separation? Is it just the Madcam1 expression due to COUPTFII overexpression? It might be interesting to speculate on the physiological consequence of Madcam1 upregulation as well in the COUPTFII overexpressing mice.

3) It appears that COUPTFII overexpression increased Madcam1 and St6gal1 expression only in certain endothelial cell types but not in others above WT levels(HEC, PCV). Why is this the case? Is it because this is maxed out expression in these cell types such that even upregulation of Nr2f2 cannot force increased Madcam1/St6gal1 expression? The authors should discuss this.

4) It appears that the lower panels in Fig. S5B are higher magnification images of the same sections in the upper panels – if this is the case, the authors should clearly mention this.

5) Is the architecture of the capillaries in the lamina propria changed in Nkx2-3 and St6gal1KOs?

6) The authors perform experiments on St6gal1^{-/-} mice in Fig.S6 – however there is no mention/discussion of these results in the text? Also, what is the difference between the upper panels in Fig.S6A and lower panels in Fig. S6B?

Figure 5:

1) It is extremely hard to read anything in fig.5B.

Overall comments:

- A more detailed explanation of the results in the text of the manuscript would be of great benefit to any reader of the paper. In the current version of the manuscript, the reader has to extensively interpret the data shown in the figures on their own. This distracts from the interesting results shown here.

- Previous work has identified Nkx2-3 sites in the Madcam1 promoter and shown their involvement through reporter assays. Is the NCCE element identified here one of the previously identified NKX sites? The authors could include this in the discussion.

- Of course, a desirable and most direct experiment would be the deletion of these NCCE's endogenously in the genome, either in an organism or maybe even in bEnd.3 cells, and the subsequent effect on the expression of the associated gene to show that these elements are directly involved in controlling tissue specific expression of certain genes.

Reviewer #3 (Remarks to the Author):

The work by Dinh et al. addresses the transcriptional regulatory logic of Madcam1 and St6gal1 that function to control lymphocyte homing into intestinal tissues, identifying a composite DNA element that coordinates inputs from the transcription factors (TFs) COUP-TFII and NKX2-3 to provide regional tissues specificity. The paper is well written and nicely integrates insights from biochemical studies on protein-DNA binding and cooperativity, in vivo TF dependence of the genes, and bioinformatics studies of the composite element genome wide. This type of detailed work to connect specific regulatory elements with genetic dependence is critical to understanding specificity in the immune system and development. This work is likely to be of interest to a broad audience interested in immune development, regulation of immune cell homing, and general transcription regulation. I have only several minor issues outlined below.

Line 104: "Overexpression of Nr2f2 alone did not active the reporter, but Nr2f2 cooperated with Nkx2-

3 to promote enhancer activity (Figure 1B)". While the bars indicate an increase in expression, there is not a significance bar indicating statistically different expression with the combination compared to Nkx2-3 alone. However, in Figure 1C, although not explicitly indicated, the combination seems to yield statistically higher levels of reporter gene activation. The authors clarify the Figure 1B situation, or perhaps use the Figure 1C data to support the conclusion to make it clearer for the reader.

Line 201: The acronym BEC does not appear to be defined.

The authors wish to thank all reviewers for their helpful comments. We address the comments point-by-point below.

Reviewer #1 (Remarks to the Author):

In this manuscript Dinh et al. address the regulation of mucosal addressin expression. They show that a composite element comprising binding sites for NKX2-3 and COUP-TFII is present in regulatory regions of *Madcam1* and *St6gal1* and functionally relevant in vitro as well as in vivo in the context of pan-endothelial overexpression in mice. Similar NCCE were found in genes regulating morpho- and organogenesis. Thus, they conclude that these NCCE are a conserved morphogenetic code directing tissue-specific vascular and immune organ specialization.

One has to congratulate the authors on this work. The study is well conducted, uses state-of-the-art methodology and the manuscript is well written. It reports significant findings with important novelty for the field. The conclusions are accurately supported by the experiments.

I have only a few comments that might help to further improve the manuscript:

1. I suggest using a color code in the bar graphs to make it easier for the reader to understand the different interventions of first view

We thank the reviewer for the suggestion and have color coded the bar graphs as requested.

2. Fig. S2A: There seems to be a dose-dependent effect of NR2F2 on human MADCAM1 expression. Can the authors further comment on that? How can that be explained? Is this similar in mice?

We saw that excess NR2F2 reduced reporter activity in both mouse and human. We hypothesize that this reflects competitive displacement of NKX2-3 by COUP-TFII at the "A" site, with COUP potentially forming homodimers. COUP-TFII homodimers are known to be repressive (Qin et al., 2014).

We removed comments about this from the manuscript to be concise but can add them back if need be.

3. Fig. 4 and associated supplemental data: Very nice data demonstrating the ectopic expression of *Madcam1* in the intestine. However, it would be important to also show data from extraintestinal tissues.

We appreciate this suggestion. In the figure below we show that *Madcam1* is ectopically expressed in iEC COUP OE mice in pancreas blood ECs, which express *Nkx2-3*. In contrast, *Madcam1* remains negative in the lung, where EC in the adult lack *Nkx2-3* (or other NKX2 family members). The figure can be added to the supplement if desirable.

A IEC CoupOE induces MAdCAM1 in BECs in pancreas but not in lung

B Pancreas but not lung BEC express *Nkx2-3* (data from Tabula Muris Senis)

- The transition to the genome-wide screening for NCCE is quite sudden and could be better explained.

We thank the reviewer for pointing that out. We have added a sentence at the beginning of the “NCCE in the genome” section for better transition (lines 249-250, highlighted).

- Some typos: e.g. l. 126, l. 489, l. 493

We thank the reviewer for pointing that out, we have fixed these.

6. citation 2 is not indicated in the reference list

This has been addressed.

We thank Reviewer #1 for these comments and have updated the text accordingly.

Reviewer #2 (Remarks to the Author):

In this manuscript, Dinh et al examine the transcriptional code involved in the venous expression of addressin molecules specifically in gut associated lymphoid tissue. Understanding the transcriptional circuitry which restricts the spatio-temporal expression of certain defining markers and in turn determines the fate of specific cell types is necessary to understand how tissue specificity arises, not only in during development, but also how it is maintained in post-natal life. As such, this work addresses the HEV tissue-specific expression of Madcam1 and St6gal1, two genes that encode addressin molecules which are important in mediating immune responses in gut associated lymphoid tissue. Using transactivation assays, the authors show that a conserved composite Nkx-CoupTF element (NCCE), found in the promoter and intron of Madcam1 and St6gal1 respectively, is required (in vitro) for directing reporter expression in 293T or b.END cells. Further using CoupTFII overexpressing mice, the authors show increased Madcam1 and St6gal1 expression in certain endothelial cell types isolated from peyer's patches of adult mice. Finally, the authors suggest that similar conserved NCCEs are found through the genome and are involved in regulating the tissue specific expression of several developmental genes.

This is an interesting paper and furthers our understanding of the tissue specific expression of addressins in GALT. Overall, I am supportive of it's publication; however I do have some concerns which if addressed would improve the quality of the paper and make it more accessible to a general audience as well. I have divided my comments based on major figures. They are as follows:

Figure 1:

1) The authors do not provide any details as to how the initial NCCE element in the Madcam1 promoter was identified – was it through specific computational analysis? Additionally, the methods section mentions ATAC-seq; however, I could not come across these data in the results section anywhere?

We deleted the mention in the methods. In early screening for regulatory sequences, we had identified that the NCCE falls within an ATAC+ peak positive in PP HEV but not PLN HEV. However as the key NCCE is in the proximal MADCAM promoter, we felt presentation of the ATAC data was not essential and removed the data to be concise.

2) The authors should explain the results of their EMSAs better in Figure S1 and not just allude cursorily to them without stating the actual result in text.

We thank the reviewer for the suggestion. We added a paragraph (lines 95-104) to elaborate on the EMSA results.

3) Is the composite element used in the reporter assays in Figure 1B and C exactly the same as the one in the mouse Madcam1 promoter?

Yes, it is the same sequence as the endogenous promoter.

4) In Fig. 1C, 293T+TNF α should read as 293+TNF α

We used 293T (SV40 T antigen expressing) not plain 293 cells in all of our experiments. 293T cells are widely used.

5) How many tandem NCCE sites were used in the reporter used in experiment for Fig.1D? Did this reporter also require additional stimulation with TNF α - a schematic would be helpful here.

We used three tandem repeats of NCCE in the reporter. We updated this information in the text (line 116) and the legend. TNF stimulation was done in the same manner across the board and indicated in the Figure 1 legend.

6) The authors mention that bEnd.3 cells express low levels of Nkx2-3, CoupTFII, and Hey1. This claim should be substantiated with either a reference or data.

We have added the raw expression data by microarray to the main text (lines 124-125) and updated the Methods section accordingly (lines 498-500).

7) Statistical analysis for the data shown in Fig. 1G are missing.

We thank the reviewer for pointing that out. The significance bars have been added.

8) It might be helpful to explain Fig.1H in a little more detail in the text.

We thank the reviewer for the suggestion and added a paragraph (lines 135-145) to provide more details for Fig 1H.

Was TNF α also used as a stimulus in this experiment?

Yes. We thank the reviewer for pointing that out, the figure has been updated.

Additionally, there is an abrogation of reporter activity in WT bEnd.3 cells which is relieved upon blocking of Hey1 in DN-MAML bEND.3 cells – does this imply that even the low level of endogenous Hey1 present in WT bEnd cells is sufficient to block activation by endogenous Nkx2-3/CoupTFII?

That is correct. It is our interpretation that overexpression of DN-MAML is relieving the Notch downstream repressors' repression of the endogenous activation complex.

9) The p values for the graphs in Fig. S2A are missing from the legend.

The figure legend has been updated.

10) In Fig. S2A, why do higher levels of Nr2f2 inhibit reporter activation in presence of Nkx2-3?

We addressed this with reviewer one, as they also had the same question. It may be due to COUP homodimers being repressive (Qin et al., 2014) and displacing the NKX:COUPTFII activation complex.

Figure 2:

1) Why is the Nkx2-3-COUPTFII heterodimeric gel shift band lower than that observed for binding by COUP-TFII alone? Is this a result of the different sizes of the hetero/homodimeric complexes formed?

COUP-TFII forms stable homodimers in solution that bind to direct repeats of the COUP-TFII half site motif with variable spacing between repeats (Qin et al., 2014). Also, COUP-TFII (45.6 kD) is larger in size than Nkx2-3 (38kD), as such, Nkx2-3:COUP-TFII heterodimer would be smaller than the homodimer.

It is also worth noting that in these non-reducing gels, migration is not well correlated at all with the MW of the complex. Each complex forms its own structure, which impacts migration sometimes.

2) The authors should explain the peptide competition assay better in the text of the results section. Additionally, the methods section is missing information on this experiment. The Y-axis in Fig. 2D should start at 0 and not at 20 (can show a broken axis if the differences are to be made more apparent)

We have updated the results (lines 177-179) and methods (lines 464-469) to better clarify the peptide competition assay. Figure 2D has been updated so that the y-axis starts at 0.

3) Why were the experiments in Fig. 2E and 2F done specifically only in DN-MAML bEND.3 cells and not in WT bEND.3 cells since Madcam1 expression can also be stimulated by TNF α and Nkx2-3 in WT cells?

Our purpose is to mimic venous EC, which lack Notch signaling. The DN MAML cells more closely reflect venous behaviour, including showing significantly higher TNF induced MAdCAM1 expression.

Figure 3:

1) The NCE in the St6gal1 locus appears to be in the 2nd Intron according to the schematic depicted in Fig. 3A and not in the first intron as mentioned in the text.

We thank the reviewer for pointing that out. We changed the text to second intron (line 190, highlighted).

2) Again, it would be helpful to explain the logic behind performing experiments only in DN-MAML bEnd.3 cells and not WT bEnd. 3 cells (as in Figs. 3C and D).

Please see above answer for Figure 2 #3.

Figure 4:

1) What is the age of mice used for the scRNA-seq experiments in Fig.4? Additionally, what is the WT genotype used for analysis? Are they pure WT mice? Tamoxifen only controls? Ideal controls would be Tamoxifen-treated Cdh5CreER tg/0 mice without the overexpressing or KO COUPTF allele since Tamoxifen+Cre induction itself is known to cause molecular changes. Either way, the authors should clearly mention what WT means in this context?

The mice are aged from 6 to 10 weeks. Littermate cre- mice treated with tamoxifen were used as controls. We mention that in the Methods section (line 507-508, highlighted).

2) It appears in the tSNE plot that there is almost a complete separation of the WT and COUPTFII-OE cells in certain clusters (HEC and CapEC) – it would be interesting to note what are the gene changes driving this separation? Is it just the Madcam1 expression due to COUPTFII overexpression? It might be interesting to speculate on the physiological consequence of Madcam1 upregulation as well in the COUPTFII overexpressing mice.

We agree that overall gene changes will be of interest and plan to focus on these broader effects of Coup in another publication. We feel that inclusion of this data here would be inappropriate as departing from our focus on the addressins and the NCCE.

3) It appears that COUPTFII overexpression increased Madcam1 and St6gal1 expression only in certain endothelial cell types but not in others above WT levels(HEC, PCV). Why is this the case? Is it because this is maxed out expression in these cell types such that even upregulation of Nr2f2 cannot force increased Madcam1/St6gal1 expression? The authors should discuss this.

We believe this is the case and have added a sentence (line 233-234) to explain this.

4) It appears that the lower panels in Fig. S5B are higher magnification images of the same sections in the upper panels – if this is the case, the authors should clearly mention this.

That is correct. We now mention this in the legend.

5) Is the architecture of the capillaries in the lamina propria changed in Nkx2-3 and St6gal1KOs?

We did not appreciate significant changes in the architecture. The architecture of vessels is likely determined by non-tissue-specific EC programs.

6) The authors perform experiments on St6gal1-/- mice in Fig.S6 – however there is no mention/discussion of these results in the text?

Thank you for that pickup. We now also cite the results in line 210-211, which are included merely to validate the SNA approach to defining St6gal1-dependent staining.

Also, what is the difference between the upper panels in Fig.S6A and lower panels in Fig. S6B?

Figure S6B also shows artery staining.

Figure 5:

1) It is extremely hard to read anything in fig.5B.

We thank the reviewer for the comment. The heatmaps in Fig 5B have been rotated and enlarged for better readability.

Overall comments:

- A more detailed explanation of the results in the text of the manuscript would be of great benefit to any reader of the paper. In the current version of the manuscript, the reader has to extensively interpret the data shown in the figures on their own. This distracts from the interesting results shown here.

We appreciate the feedback for improvement. We have added more details to describe and interpret the results.

- Previous work has identified Nkx2-3 sites in the Madcam1 promoter and shown their involvement through reporter assays. Is the NCCE element identified here one of the previously identified NKX sites? The authors could include this in the discussion.

Yes, the NKX site in the Madcam1 NCCE was previously evaluated in fibroblasts (Pabst et al., 2000). This study is cited in our paper.

- Of course, a desirable and most direct experiment would be the deletion of these NCCE's endogenously in the genome, either in an organism or maybe even in bEnd.3 cells, and the subsequent effect on the expression of the associated gene to show that these elements are directly involved in controlling tissue specific expression of certain genes.

We of course agree. However, in our studies we do assess the effects of multiple specific NCCE mutations and deletions in the controlled context of reporter expression in bEnd.3 EC.

We thank Reviewer #2 for these comments and have updated the text accordingly.

Reviewer #3 (Remarks to the Author):

The work by Dinh et al. addresses the transcriptional regulatory logic of Madcam1 and St6gal1 that function to control lymphocyte homing into intestinal tissues, identifying a composite DNA element that coordinates inputs from the transcription factors (TFs) COUP-TFII and NKX2-3 to provide regional tissue specificity. The paper is well written and nicely integrates insights from biochemical studies on protein-DNA binding and cooperativity, in vivo TF dependence of the genes, and bioinformatics studies of the composite element genome wide. This type of detailed work to connect specific regulatory elements with genetic dependence is critical to understanding specificity in the immune system and development. This work is likely to be of interest to a broad audience interested in immune development, regulation of immune cell homing, and general transcription regulation. I have only several minor issues outlined below.

1. Line 104: "Overexpression of Nr2f2 alone did not activate the reporter, but Nr2f2 cooperated with Nkx2-3 to promote enhancer activity (Figure 1B)". While the bars indicate an increase in expression, there is not a significance bar indicating statistically different expression with the combination compared to Nkx2-3 alone. However, in Figure 1C, although not explicitly indicated, the combination seems to yield statistically higher levels of reporter gene activation. The authors clarify the Figure 1B situation, or perhaps use the Figure 1C data to support the conclusion to make it clearer for the reader.

The comparison between Nkx2-3+CoupTFII vs Nkx2-3 alone is marginal in 1B ($p < 0.1$) but is significant in 1C. We thank the reviewer for the suggestion and have added the significance bar in Figure 1B and cited Figure 1C in the text (lines 112-113).

2. Line 201: The acronym BEC does not appear to be defined.

We thank the reviewer for pointing that out. The acronym "BEC" has been replaced with "blood EC" since it has only been referred to twice in the text (lines 220 and 222).

We thank Reviewer #3 for these comments and have updated the text accordingly.

REVIEWERS' COMMENTS

Reviewer #1 (Remarks to the Author):

The authors have adequately addressed the reviewers' comments. I would welcome the figure provided in the point-by-point reply to be included as a Supplemental Figure.

Reviewer #2 (Remarks to the Author):

The authors have addressed all my concerns and I am supportive of this manuscript's publication.

Reviewer #3 (Remarks to the Author):

I am satisfied that the authors addressed my concerns.